# Identification of Multi-Class Drugs Based on Near Infrared Spectroscopy and Bidirectional Generative Adversarial Networks

**DOI:** 10.3390/s21041088

**Published:** 2021-02-05

**Authors:** Anbing Zheng, Huihua Yang, Xipeng Pan, Lihui Yin, Yanchun Feng

**Affiliations:** 1School of Automation, Beijing University of Posts and Telecommunications, 10 Xitucheng Road, Haidian District, Beijing 100086, China; spztf@bupt.edu.cn; 2School of Computer Science and Information Security, Guilin University of Electronic Technology, No.1 Jinji Road, Qixing District, Guilin 541004, China; pxp201@guet.edu.cn; 3China Institute for Food and Drug Control, 2 Tiantan Xili, Dongcheng District, Beijing 100086, China; yinlihui@nifdc.org.cn (L.Y.); fyc@nifdc.org.cn (Y.F.)

**Keywords:** near-infrared spectroscopy, drug identification, multi-class classification, deep learning, generative adversarial networks

## Abstract

Drug detection and identification technology are of great significance in drug supervision and management. To determine the exact source of drugs, it is often necessary to directly identify multiple varieties of drugs produced by multiple manufacturers. Near-infrared spectroscopy (NIR) combined with chemometrics is generally used in these cases. However, existing NIR classification modeling methods have great limitations in dealing with a large number of categories and spectra, especially under the premise of insufficient samples, unbalanced samples, and sensitive identification error cost. Therefore, this paper proposes a NIR multi-classification modeling method based on a modified Bidirectional Generative Adversarial Networks (Bi-GAN). It makes full utilization of the powerful feature extraction ability and good sample generation quality of Bi-GAN and uses the generated samples with obvious features, an equal number between classes, and a sufficient number within classes to replace the unbalanced and insufficient real samples in the courses of spectral classification. 1721 samples of four kinds of drugs produced by 29 manufacturers were used as experimental materials, and the results demonstrate that this method is superior to other comparative methods in drug NIR classification scenarios, and the optimal accuracy rate is even more than 99% under ideal conditions.

## 1. Introduction

In the drug market, different drugs and different brands will have different pricing. Sellers can use fake packaging on low-cost pharmaceutical products and sell them as high-priced drugs. They may also use inferior brand drugs of the same drug as famous brand products to sell at high prices in the market. Therefore, it is of great significance in drug supervision to identify the true source of drugs by classification and identification of multiple drugs produced by multiple manufacturers.

Near-infrared spectroscopy (NIR) has the advantages of low instrument cost, direct measurement, non-destructive detection, and on-site detection, which is suitable for rapid qualitative and quantitative analysis of drugs [1,2,3]. It is usually combined with chemometrics methods such as partial least squares discriminant analysis (PLS-DA) [3,4,5], linear support vector machine (Linear SVM), and other linear classifiers [6,7,8,9,10] and BP-ANN classifier [10,11] in a classification scenario. In recent years, some deep learning methods, such as stack sparse auto-coding (SAE) [12], deep belief network (DBN) [13], deep convolution neural network (CNN) [14], have also been reported in drug identification and classification modeling.

Classification and identification are also very important in the analysis and processing scenarios of sensors’ collected data, so the above methods are often used in other sensor data analysis scenarios, not only in the chemometrics domain or NIR domain, and the following problems of NIR classification and identification are also common in the scene of sensor data acquisition, analysis, and processing.

Due to the famous “curse of dimensionality” problem [15] in classification methods and the low quantitative detection limit of near-infrared spectroscopy [2], the combination of traditional chemometrics and near-infrared spectroscopy is not suitable for scenes with a large number of categories and spectra. Meanwhile, the classifiers of deep learning methods are suitable but often need enough samples to participate in training to achieve the desired effect. Thus, both traditional chemometrics methods and deep learning methods require sufficient samples within the class and balanced samples between classes in modeling. 

However, in the practice of drug inspection, the number of drug varieties on the market is large, usually involving a large number of categories and a large number of spectra. At the same time, the spectrum samples available are usually unbalanced, and the number of spectra within each category is usually a long-tail trend, that is, the number of intra-class spectra in a few categories is particularly large, while the number of intra-class spectra of more categories is often insufficient.

This will lead to two important adverse consequences: “false high” of accuracy and uneven distribution of model error. 

Due to a large number of classes are insufficient in samples, which leads to inadequate extraction of class features, resulting in the low classification accuracy of these classes. However, since the test proportions of them are also low, the urgency of their lower accuracy will never be exposed compared with the dominantly large classes’ high accuracy. The accuracy rate of the whole classification may be very high, but for some of the important classes, their accuracies are even too low to be used at all. 

Besides, for the finished classifier, due to the imbalance of the number of spectra between classes, the distribution of classification errors is also uneven. Most of the misclassified spectra are classified into the categories with fewer spectra within the class, while the categories with more spectra are very few. Generally, the classes with more intra-class spectra are drugs that are easy to collect samples, such as cheap drugs and common drugs, while those with fewer intra-class spectra are rare drugs that are not easy to obtain samples, such as expensive drugs or newly developed special drugs. If the classifier often classifies common drugs into rare drugs, the consequences are often serious, which means that cheaters cheat successfully, the interests of consumers are greatly damaged, and even the lives of patients are endangered. The classification of rare drugs as common drugs is not so important, because the owners of rare drugs often ask for further inspection and provide other more reliable evidence so that the correct results can always be obtained.

Although the most fundamental way to solve the above problems is to collect more samples, it is usually conditional and expensive, and it is difficult to ensure that a complete, reliable, and sufficient number of samples are collected for specific problems. Under this premise, the second method, which uses high-quality and diversified sample generation methods, generates samples on demand [16] for classification, becomes the most worthy method to be considered.

In recent years, the generative adversarial networks [17] (GAN) method, which is popular in the field of deep learning, is just in line with this idea. Since the original GAN was reported in 2014, it has experienced a long time of rapid development and has evolved many practical and mature sample generation methods [18], such as conditional GAN (C-GAN) [19,20], deep convolutional GAN (DC-GAN) [21], bidirectional GAN (Bi-GAN) [22], cycle GAN [23], etc. In the fields of images [24,25,26,27], videos [28,29,30,31,32], voices [33,34,35,36,37,38] and even natural languages [39,40,41,42], excellent sample generation effects have been achieved. “DeepFake” programs [43,44,45] based on a modified GAN can even generate synthetic artificial faces that can’t be distinguished from real ones by both humans and machines, which leads to the discussion of artificial intelligence ethics because of its excellent sample generations.

Unfortunately, in the field of near-infrared spectrum detection (including the field of drug near-infrared spectrum detection), and even in the wider field of sensor data processing, there is no relevant report on the application of GAN methods. Therefore, we can only modify an appropriate GAN method that is most suitable for the feature extraction and the generation of category samples on demand. The candidate methods are Info-GAN [46], Bi-GAN, VAE-GAN [47], etc. according to our experience and the implementation difficulties, we finally select Bi-GAN as the modification object to achieve the goal of this paper.

Based on this background, this paper constructs a multi-classification model of drugs based on near-infrared spectroscopy and Bi-GAN sample generation, so that it can correctly classify in the scene of a large number of categories and spectra, and effectively solve the problems of insufficient samples, unbalanced samples, and cost-sensitive classification errors in the classification process.

In the scenario of a large number of categories and a large number of spectra, the problems of insufficient samples, unbalanced samples, and cost sensitivity of classification errors are common in other sensor data classification tasks, so this method can also be applied to data classification tasks obtained by other sensors.

## 2. Materials and Methods

### 2.1. Materials

All the materials used in this paper were obtained from the China Institute for Food and Drug Control (Beijing, China). A total of 1721 samples of four drugs (metformin hydrochloride tablets, chlorpromazine hydrochloride tablets, chlorphenamine maleate tablets, cefuroxime axetil tablets) produced by 29 manufacturers were collected. All samples were measured by FTIR spectroscopy (Matrix F spectrometer, Bruker Corporation, Billerica, MA, USA). Before sample collection, the instrument passed a self-diagnosis test and calibration. The wavelength range of data is 4000–11,995 cm^−1^, and the resolution is 4 cm^−1^.

The near-infrared spectra of the drugs were recorded using a diffuse reflection optical fiber probe. SMA 905 standard interfaces were used for coupling the optical fiber, light source and spectrometer. The ambient temperature was 18–30 °C, and the air humidity was less than 70%. All samples used the same determination background. The measurement operation followed a unified operation protocol, as shown in Figure 1.

Sample information is shown in Table 1. It can be seen from this table that the number of samples is not balanced. Some classes have more samples, reaching 135, while others have fewer samples, only 21. The sample numbers within the class are sorted from high to low to form a column chart, as shown in Figure 2, and the distribution histogram is shown in Figure 3.

As can be seen from Figure 2 and Figure 3, most of the samples in the graph are concentrated in the top eight categories, accounting for 46.31% of the weight, and the following 21 categories only represent 53.69% of the weight. For more than half of the classes, the number of samples ranged from 21 to 54, which was less than the average sample size of 59 and far less than the highest sample number of 135. Therefore, it can be asserted that the number of samples in this dataset is extremely unbalanced, and the samples in many categories are insufficient.

The spectra of four kinds of drugs produced by various manufacturers are shown in Figure 4. As can be seen from the figure, the spectra of the same drug produced by different manufacturers according to the Pharmacopoeia of the People’s Republic of China (2015 version II) are very similar, and the important bands (peak and valley positions) mostly overlap. Metformin hydrochloride tablets manufactured by two manufacturers (No. 6 and 7) and chlorphenamine maleate tablets manufactured by two manufacturers (No. 18 and 19) were taken from Figure 4 to form Figure 5, and the difference between manufacturers of the spectra could hardly be detected by visual inspection.

Generally, the time from R&D to final registration of the original drug is about 15 years, and it needs to undergo four-phase clinical trials at a cost of hundreds of millions of dollars. Such drugs cannot be imitated until the patent has expired, and the enjoy the protection of policies such as separate pricing. Generic drugs only replicate the main components of the original drug, and even if a huge investment is invested in the generic process, the price is only about 1/3 even 1/6 that of the original drug. Therefore, it is understandable that generic drugs and the original drug can be as consistent as possible without being distinguished.

This is very challenging for classification algorithm modeling. It requires that the algorithm be able to distinguish subtle differences between the classification features when extracting the class features.

### 2.2. Methods

Under the above severe classification requirements, we build a classifier based on Bi-GAN generating sample method to achieve fair and accurate classification.

Its main idea is to use artificial a generative adversarial network to generate samples to supplement and improve the sorting of the original samples as shown in Figure 6. Through a fair and reasonable sampling strategy, each category can get enough attention in the model construction, and finally effectively alleviate the shortcomings caused by insufficient intra-class samples and unbalanced inter-class samples in the drug near-infrared spectrum classification method, making the cost of wrong classification problem can be solved effectively. 

The key to its realization lies in the modification of the original Bi-GAN. On the one hand, to make original Bi-GAN have the ability to generate specific class samples instead of random “real” samples, and the other is to make classification supervision run through every process of Bi-GAN training, let the training of generator and discriminator be interfered by the classification loss.

#### 2.2.1. Original Bi-GAN

The internal structure of the original Bi-GAN is shown in Figure 7.

Its main objectives are shown in Equation (1):(1)$$\underset{G,E}{\min}\underset{D}{\max}V(D,E,G)$$
where:(2)$$V(D,E,G):={Ε}_{x\sim p_{x}}[\underset{\log D(x,E(x))}{\underset{︸}{E_{z\sim p_{E}}{}_{\left(\cdot |x\right)}[\log D(x,z)]}}]+E_{z\sim p_{z}}[\underset{\log(1-D(G(z),z))}{\underset{︸}{E_{x\sim p_{G}(\cdot |z)}[log(1-D(x,z))]}}]$$

In Equations (1) and (2), *G* is the generator, which can be regarded as the decoder. *D* as the discriminator and *E* as the encoder. *x* represents the real sample. *E*(*x*) represents the representation encoded into the potential space, and it is also the extracted features. *z* is the random sampling of the prior distribution, and *G*(*z*) represents the sample generated by *z*. *y* is the data source, if the data to be discriminated comes from the real sample *x*, then *y* = 1; if it comes from the generated sample *G*(*z*), then *y* = 0.

Equation (2) shows that Bi-GAN binds the original spectrum *x* and its extracted feature *E*(*x*), and the generated sample *G*(*z*) is bound with its prior distribution sample *z*, and then the two couples is been labeled with 1 and 0 respectively. The discriminator *D* is required to distinguish them to the maximum extent, and the generator *G* is required to prevent discriminator *D* from distinguishing. After training *D* and *G* alternately, generator *G* and discriminator *D* reach a Nash equilibrium. At this time, it can be considered that the authenticity of the generated samples has little difference from the “REAL” samples, and *G* has become a usable sample generator. The effectiveness of the above-mentioned methods in sample generation and feature extraction has been confirmed in reference [22].

However, the prior distribution sampling of the original Bi-GAN is usually the random sampling of the standard normal distribution ***N***(0,1), and the category of the generated sample *G*(*z*) is not guaranteed, so it is impossible to determine whether the generated sample is the sample of the specified class. This is not in line with our goal of generating a specific class of samples. Therefore, we need to modify the original Bi-GAN to ensure the generator *G* can generate demanded random samples of “specified categories”.

#### 2.2.2. The Modifications of Original Bi-GAN

The overall modified design based on the original Bi-GAN is shown in Figure 8.

In Figure 8, we made the following changes to Bi-GAN:(1)The sampling of *z* is limited.

We limit the sampling of *z* and set the mean and variance of *P*(*E*(*x_i_*)) as shown in Formula (3). The default values of *σ* all set to 1 at first, and then they are automatically adjusted according to the previous five history records during the training process:(3)$$\left\{ \begin{array}{l} \mu_{i}=E(x_{i})\\ \sigma_{i}=\sqrt{\frac{\sum_{j=i-5}^{i-1}{\left(E(x_{j})-\mu_{i}\right)}^{2}}{4}} \end{array}\right.$$
when generating the spectrum, the real spectral template *x_i_* must be specified, and then its class label *c_i_* is also determined. *x_i_* is encoded into *E*(*x_i_*) by encoder *E*, and then the mean and variance of the prior normal distribution *P*(*E*(*x_i_*)) is determined according to Formula (3), where the feature vector *z* could be randomly sampled in the fixed mean and local average variance scope.

(2)Limit the random *G*(*z*) in a specified class.

We do this by building a classifier. The classifier *C* in this paper is composed of MLP and softmax. In the pre-training, the real sample *x_i_* is used as its input, and the generated sample *G*(*z*) is used as its input in the formal training. Its output is a predicted class *c_i_*.

The classifier should be pre-trained, and its loss function shown as Equation (4):(4)$$Loss_{classfication}=-\sum_{i=1}^{k}\left(c_{i}\cdot \log({\hat{c}}_{i}\right))$$
where *c_i_* is the class label of the real sample *x_i_* in pre-training, and for the generated sample, the class label of its spectral template *x_i_* is taken. *k* is the total number of drug categories and ${\hat{c}}_{i}$ is the predicted category. 

*G*, *D*, and *C* are alternately optimized by gradient descent algorithm, and the optimization objective is changed to:(5)$$\underset{G,E,C}{\min}\underset{D}{\max}V(D,E,G,C)$$
where:(6)$$V(D,E,G,C):={Ε}_{x\sim p_{x}}[\underset{\log D(x,E(x)}{\underset{︸}{E_{z\sim p_{E}}{}_{\left(\cdot |x\right)}[\log D(x,z)]}}]+E_{z\sim p_{z}}[\underset{\log1-D(G(z),z)}{\underset{︸}{E_{x\sim p_{G}(\cdot |z)}[(1-\log D(x,z))]}}]+E_{z\sim p_{z}}[\underset{\log C(G(z))}{\underset{︸}{E_{y\sim p_{C}(\cdot |G(z))}[\log C(G(z))}}]$$

According to this objective, the loss of discriminator during training is calculated as Equation (7):(7)$$Loss_{discrimination}=-\sum_{i=1}^{n}\left(y_{i}\cdot \log({\hat{y}}_{i})+(1-y_{i})\cdot \log(1-{\hat{y}}_{i})\right)+Loss_{classification}$$
where *y**_i_* represents the data source. If the data to be identified comes from a real sample, then *y**_i_* = 1; if it comes from the generated sample, then *y**_i_* = 0. ${\hat{y}}_{i}$ is the discriminator’s prediction, and the *Loss_classification_* is the result of Equation (4).

The loss of generator *G* during training is calculated using Equation (8):(8)$$Loss_{generation}=-\sum_{i=1}^{n}\log{\hat{y}}_{i}+Loss_{classification}$$

In this way, the classifier involves all *G*, *D* training processing. During the iterations, the generator will increasingly tend to generate samples of the same class as the template spectra.

#### 2.2.3. Sampling Strategy in Data Set Processing

In this paper, after the spectra of each category are divided into the training set and test set according to the chosen proportion, they do not directly participate in the training and testing except for the pre-training of classifiers. Instead, the number of spectra of each category participating in the training and testing is determined in an equally fixed number, and they are extracted from data set by random sampling with replacement method.

The advantages of this method are:

Firstly, each class’s participating spectra in the training course are equal, so equal attention can be paid to each category, and the categories with fewer samples in the training process will not be ignored.

Secondly, even for the same spectrum template, because there is a random sampling process in the generation phase, the final generated spectrum will be different, so the diversity is guaranteed to a certain extent.

#### 2.2.4. Application of Classifier

After the training of Bi-GAN, three trained networks, *E*, *G*, and *C*, are taken out to form a structure as shown in Figure 9, which is used to predict the categories.

For each true spectrum *x* to be predicted, we can repeat the input a fixed number of times. Due to the existence of *P*(*E*(*x*)), the model will produce a different synthetic spectrum each time, which is consistent with the real spectrum in their categories but has good diversity. Most of the prediction results should be consistent with the category of *x* except for one or two abnormal values. By counting the frequency of output results, we can select the category with the highest frequency, which can be decided as the final prediction result of the *x*.

In this way, when *P*(*E*(*x*)) sampling occasionally appears small probability sampling anomaly, the model will not be disturbed by it, and finally, the correct category is selected.

## 3. Results

### 3.1. Experimental Environment

This paper uses the following hardware and software environment for the data modeling experiments:

Hardware environment: CPU Xeon 2678v3 (12 cores, 24 threads), memory 64 GB, SSD 1TB, GPU NVIDIA Tesla V100.

Software environment: operating system Ubuntu 18.04.3 LTS, NVIDIA driver version 440.33.01, CUDA v10.2, cudnn v7.6.5, keras GPU 2.3.1, tensorflow GPU 1.15.0, sci-kit learn 0.19.0.

### 3.2. Muti-Classification Results

In the experiment, E, G, and C are constructed by multi-layer perception (MLP):

E network MLP uses 2074-120-30 to set up the network, layer 120 and layer 30 are preceded by dropout (0.2), followed by batch normalization layer (BN) decoding. The activation function is RELU.

G network MLP uses 30-360-2074 to set up the network, and the activation function is also RELU.

The C network classifier is designed with 2074-150-30-softmax, and the activation function is sigmoid.

All the networks use RMSprop optimizer, and its parameters are the Keras’ default parameters, the batch size is set to 60, trained for 150 epochs.

The experimental spectra were divided into a training set and test set according to 9:1, 8:2, 7:3, 6:4, 5:5, 4:6, 3:7, and 2:8. 

The training set is used for modeling, and the test set is used to verify the effectiveness of the model. For example, in the first row of Table 1, for the metaformin hydrochloride tablets produced by Shanghai Xinyi Pharmaceutical Factory Co., Ltd. (Zhengzhou, China), if the training set and test set are divided by 9:1 into 94 samples, 89 samples are randomly selected to be put in the training set for modeling, and the remaining nine samples are put in the test set for verifying the effectiveness of the model. The nine samples in the test set are invisible during the modeling period. They are “non-existent” external samples for the training process, but they are internal samples for the whole data set because their distribution and internal properties are similar to those of the 89 samples participating in the training.

Each experiment was conducted 10 times and the best results were recorded. The experimental results are shown in Table 2.

As can be seen from Table 2, when the training set’s proportion is more than 50%, the classification accuracy rate of the multi-classification model in this paper is more than 99% and when the proportion decreases, the classification accuracy does not decrease accordingly before 4:6, as if it has little relationship with the division of training set and test set, but when the training set is only 40% of the total, the accuracy displays a big drop from 99% to 92%. Since most of the categories in the data set have 30–50 spectra, when the training set accounts for less than 40%, most of the categories obviously begin to reflect missing data. When the training set proportion is only 30%, there are only 6 spectra of the minimum category that can be used for training. While when the training set reaches 20%, there are only four spectra of the minimum category that can participate in the training, and most categories (15 out of 29 categories) have only 10 or fewer data pieces for training.

To investigate the classification errors of each category, we draw the confusion matrix in the case of the most favorable classification (90% of the training set), as shown in Figure 10, and the confusion matrix in the case of the worst classification (20% of the training set), as shown in Figure 11.

From Figure 10, it can be seen that in the case of the most favorable classification (90% of the training set), except for categories 1, 3, and 9 (for tools reasons, the figure count classes from 0, while the information table count categories from 1, we apologize for the inconvenience), all the classes classified perfectly.

The intra-class spectra number of categories 1, 3, and 9 are 94, 67, and 48. Only one category falls into the insufficient data interval 21–54 of Figure 2. However, Category 4, which has the least number of spectra within its class, has a good classification effect. This shows that even if the classification error occurs in this situation, the error is not caused by the lack of intra-class spectra.

As can be seen from Figure 11, those classes with lower classification accuracy are more evenly scattered in the intervals shown in Figure 2, but not in the areas with insufficient data. This shows that the method in this paper has played a due role in eliminating the adverse effect of insufficient spectral numbers in the class.

It can also be seen from Figure 11 that the classification errors are mainly caused by the misclassification among different manufacturers within the same drug. Besides, whether it is in Figure 9 or Figure 10, the classification accuracy of Categories 25–29 (corresponding to cefuroxime axetil tablets) is almost not affected by the decrease of training spectrum proportion. It can be seen that in this method, the most important factor affecting the accuracy of classification is still the inherent characteristics of the spectrum of various pharmaceutical products, which is consistent with our original classification purpose.

### 3.3. Comparative Methods Results

The experimental results are compared with three kinds of algorithms: one is the traditional linear classification algorithm, mainly PLS-DA and linear SVM; the other is the traditional nonlinear classification algorithm, mainly RBF SVM, k-NN, BP-ANN; the third is the deep learning algorithm in recent years, mainly DBN, SAE, and CNN. Among them:

The number of components of PLS-DA is the same as that of *z* in this paper, which is set to 30. The *c* value of linear SVM is 1. 

The k-NN’s k is set to 1. The RBF SVM’s Gamma value is set to 0.0001, and the *c* value is set to 1. The BP-ANN takes two layers, the number of units in each layer is set to 2074-29, and the activation function is sigmoid.

In DBN, only one layer of RBM (1037 units) is set, followed by a full connection layer and a softmax classifier as the output.

The SAE’s codec takes two layers, the number of units in each layer is set to 2074-180-30 and 30-180-2074, respectively. The feature layer (30 units) is fully connected to a softmax classifier as the output.

The CNN is constructed according to the optimal method design of the reference [15].

After all the models are well trained, we run the same tests according to our mothed proposed in this paper. The comparison accuracy results are shown in Table 3.

It can be seen from Table 3 that:

Overall, the accuracy of the Bi-GAN classifier is better in accuracy than the others. DBN, SAE, CNN, and other deep learning algorithms take second place, PLS-DA and linear SVM still available since the traditional linear classification algorithm also has a certain effect in discriminating the composition of drugs.

Except for PLS-DA, when the partition of the training set and test set is extreme, each algorithm will encounter the inflection point of classification accuracy when encountering the lack of necessary class data. Among them, the sensitivity of the nonlinear algorithms is higher than that of the linear algorithm.

In the drug multi-classification algorithms, the old PLS-DA algorithm still has good performance, and it is still worthy of attention when only focusing on the influence of drug components without considering the nonlinear influence factors.

Although Bi-GAN has high accuracy, it has the highest sensitivity in data missing. Once the data missing is serious, it is easy to deviate. The training time and inferring time of the algorithms are shown in Table 4.

It can be seen from the table that:

Except for Bi-GAN, the training time of all algorithms decreases with the decreasing of training set proportion, while the inferring time shows an upward trend because test sets are expanding.

The k-NN algorithm has the least training time, but its accuracy is the worst, and its inferring time is longer than most of the others. 

Although PLS-DA, linear SVM, RBF-SVM, k-NN, and BP-ANN use CPU to calculate, their training can be completed in less than 1 s because of their simple structure. But, the inferring time of linear SVM and RBF-SVM is longer than that of nonlinear algorithms, including the deep learning algorithms.

The deep learning algorithms’ training time is longer than that of linear algorithms, but their inference time is shorter.

Among the deep learning algorithms, the cost of training time and inference time of our method is average. Its cost is fixed and does not vary greatly with the division of training set and test set, and that is a merit.

PLS-DA, Linear SVM, RBF-SVM, k-NN, and BP-ANN use scikit-learn software to train and test, scikit-learn uses the CPU for calculation and only uses a single core and a single thread.

## 4. Discussion

Construction of the NIR classification model for multi-variety and multi-manufacturer drugs involves much data and complex categories and the application scenarios are challenging. This paper starts with the analysis of the problems of insufficient samples within the class and unbalanced samples between classes in the near-infrared spectrum classification of drugs. Then analyze the cost-sensitive problems of the incorrect classification caused by these problems. Through the modified Bi-GAN, the quantitative generated samples are used instead of the original uneven real samples as the classification training basis, which can effectively solve these problems above to an extent.

By constructing the appropriate network connection, using the appropriate combination of cost functions and a fair sampling strategy, we have achieved excellent classification results in the experiments. The experimental results demonstrate that in this scenario, the proposed method can achieve a classification accuracy of more than 99% in most cases where the training set accounts for more than 50% of the whole data set. Moreover, although the accuracy of this method will be greatly reduced when the proportion of the training set is reduced to 40%, the classification accuracies are relatively stable before that. As for time cost, although the training and inferring time cost of this method are at an average level compared with other deep learning methods, its cost is relatively constant and it also does not fluctuate with the increase or decrease of the number of samples of the dataset.

By comparing the traditional and three new kinds of drug classification algorithms, we can assert that this method has successfully achieved our expectation by solving the pre-set problems, the accuracy and stability of the method for the identification of multi-class drugs by near-infrared spectroscopy are also improved to a certain extent, which can provide a useful reference in the similar scene of near-infrared spectrum analysis and sensor signal data processing.

## 5. Conclusions

We propose an improved Bi-Gan method to classify the near-infrared spectra of drugs given the problem of insufficient samples within the class and imbalance of samples between classes. By limiting the mean and variance of latent variables, adding the classification loss constraint, and using the fair strategy of sampling with replacement. We achieve the desired results. The experimental results showed that the best classification accuracy of 1721 NIR spectra of four kinds of drugs produced by 29 manufacturers was significantly 99.4%. Compare to the other eight NIR multi-classification methods in recent years, this method has obvious advantages.

The problems in this paper may also exist in other sensor data classification processing, so the method we propose can be a useful reference for readers in dealing with the multi-classification problems in other scenarios.

## Figures and Tables

**Figure 1 sensors-21-01088-f001:**
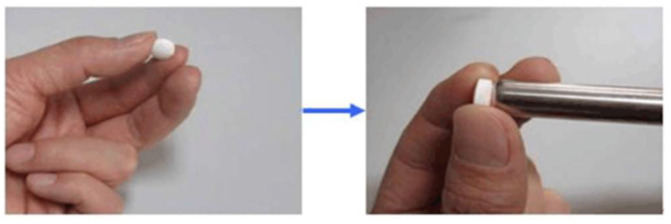
Operation method for spectrometric measurement of ordinary tablets.

**Figure 2 sensors-21-01088-f002:**
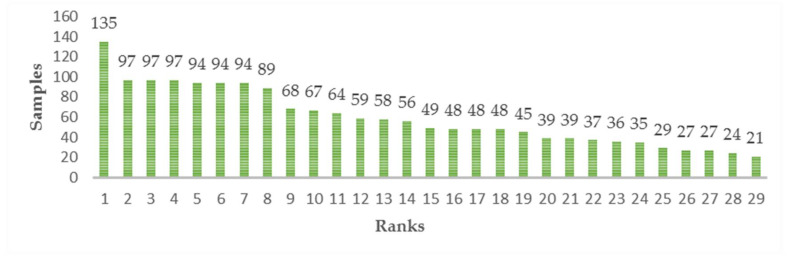
Intra-class samples are sorted from high to low. Every column stands for a class, the height value is the intra-class sample counts.

**Figure 3 sensors-21-01088-f003:**
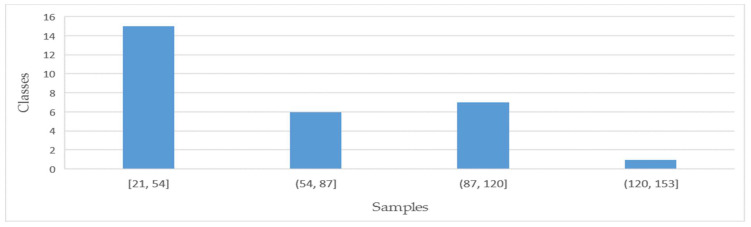
Intra-class samples distribution. Each column represents a range of samples, and the height value is the count of classes belonging to this range.

**Figure 4 sensors-21-01088-f004:**
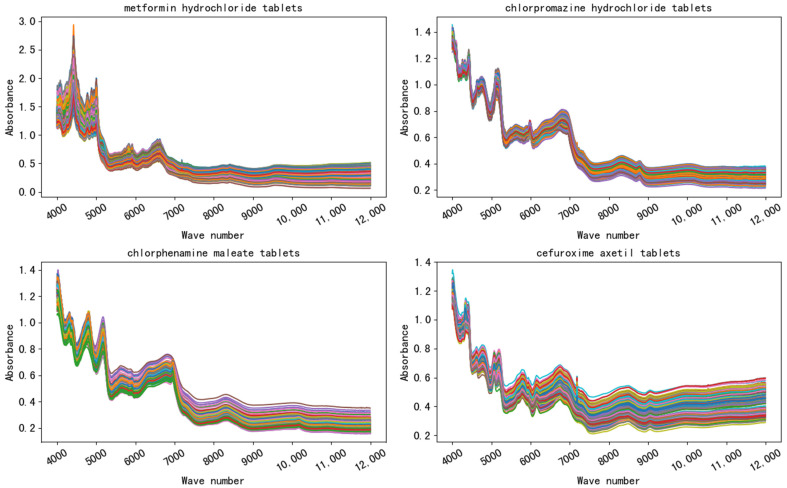
The spectra of four drugs. The same drug produced by different manufacturers according to the Pharmacopoeia of the People’s Republic of China (2015 version II) are very similar, and the important bands (peak and valley positions) mostly overlap.

**Figure 5 sensors-21-01088-f005:**
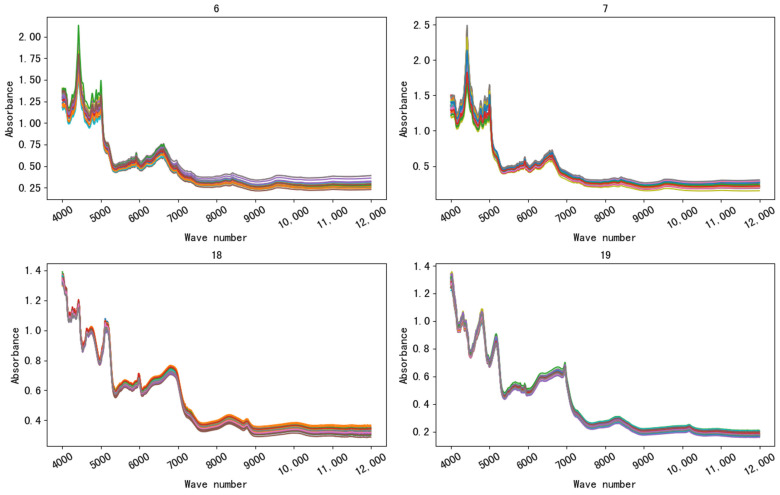
Similar spectra of the same drug produced by different manufacturers. Metformin hydrochloride tablets came from Nos. 6 and 7. Chlorphenamine maleate tablets came from Nos. 18 and 19.

**Figure 6 sensors-21-01088-f006:**
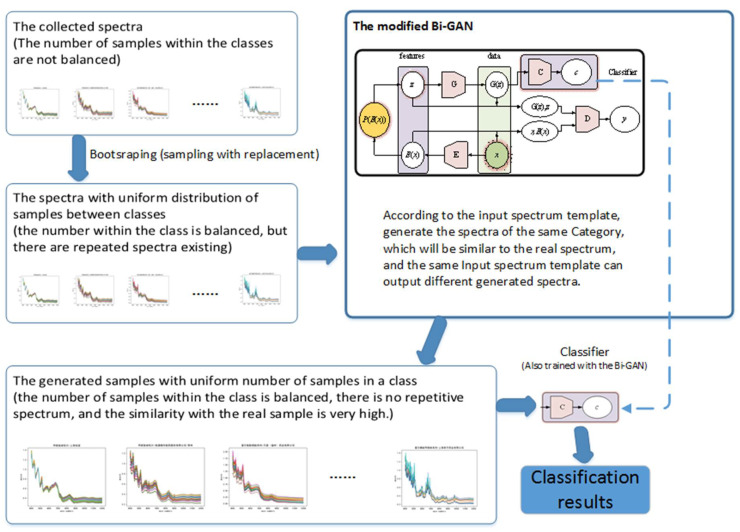
The main process of the proposed method.

**Figure 7 sensors-21-01088-f007:**
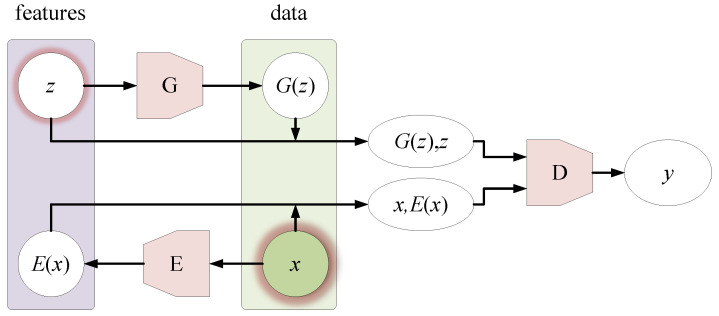
The internal structure of the original Bi-GAN.

**Figure 8 sensors-21-01088-f008:**
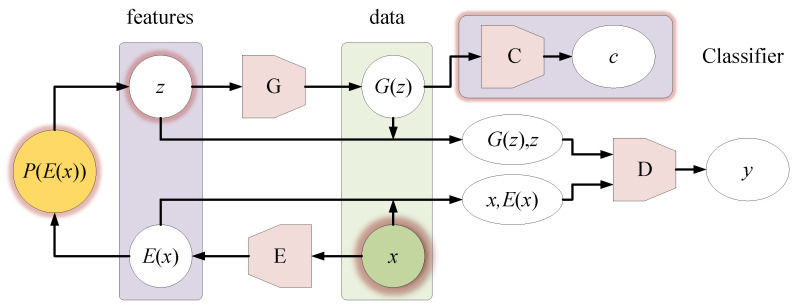
The modification of the original Bi-GAN. This is also the main design of this paper.

**Figure 9 sensors-21-01088-f009:**
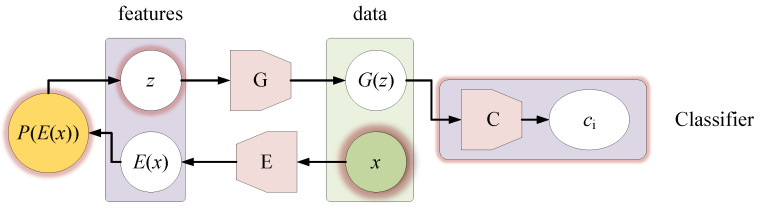
The model structure for practical application. Just input the spectrum to be classified at *x*, and the predicted category *c_i_* will be output.

**Figure 10 sensors-21-01088-f010:**
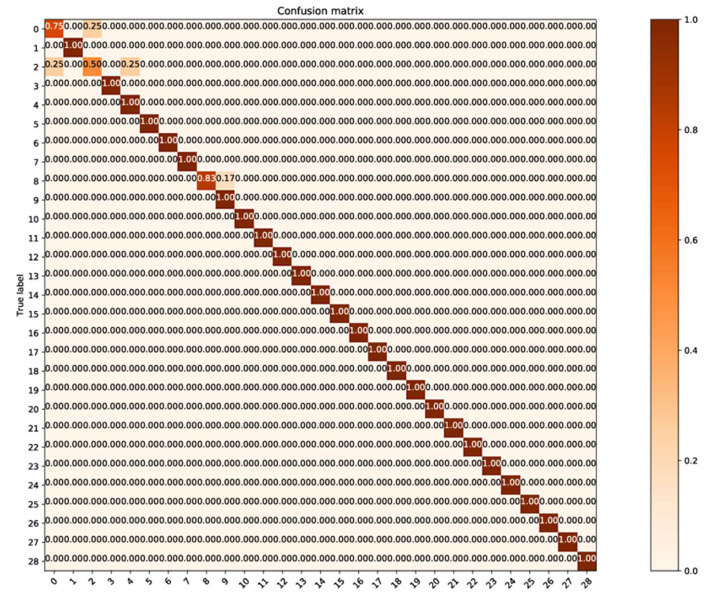
The confusion matrix in the case of the most favorable classification (90% of the training set).

**Figure 11 sensors-21-01088-f011:**
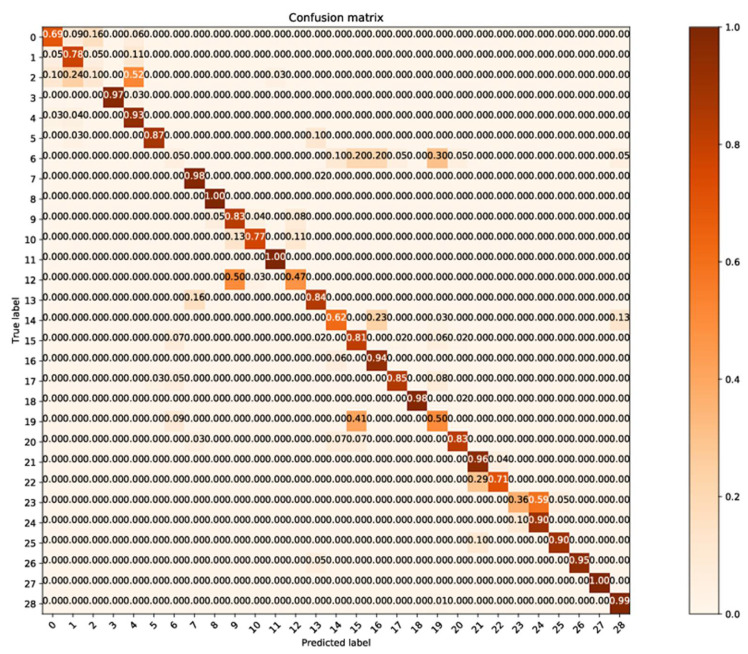
The confusion matrix in the case of the worst classification (10% of the training set).

**Table 1 sensors-21-01088-t001:** Drug Names, Manufacturers, Sample Names of the material used in this paper.

No.	Drug Name	Manufacturer	Samples
1	metformin hydrochloride tablets	Xinyi Pharmaceutical Factory Co., Ltd., Shanghai, China	94
2		Hengshan Pharmaceutical Co., Ltd. Shanghai, China	48
3		Zhonghui Pharmaceutical Co., Ltd., Beijing, China	67
4		Yongkang Pharmaceutical Co., Ltd., Beijing, China	21
5		Pacific Pharmaceutical Co., Ltd., Tianjin, China	48
6		Chuanli Pharmaceutical Co., Ltd., Chengdu, China	64
7		Baiyunshan Tangyin Dongtai Pharmaceutical Co., Ltd., Guangzhou, China	27
8		Qilu Pharmaceutical Co., Ltd. Ji’nan, China	35
9		Suzhong pharmaceutical group Co., Ltd., Taizhou, China	48
10		Jingfeng Pharmaceutical group Co., Ltd., Beijing, China	24
11		Zhonglian Pharmaceutical Co., Ltd., Shenzhen, China	68
12		Zhongxin Pharmaceutical Group Co., Ltd., Tianjin, China	97
13		Yabao Pharmaceutical Technology Co., Ltd., Tianjin, China	97
14		Kangrui Pharmaceutical Co., Ltd., Tianjin, China	97
15	chlorpromazine hydrochloride tablets	Xinyi Jiufu Pharmaceutical Co., Ltd., Shanghai, China	59
16		Yungang Pharmaceutical Co., Ltd., Datong, China	94
17		Changzhou Compass Pharmaceutical Co., Ltd., Changzhou, China	58
18		Guangdong Petty Pharmaceutical Co., Ltd., Guangzhou, China	135
19		Jiangsu Tianshili Diyi Pharmaceutical Co., Ltd., Huai’an, China	49
20	chlorphenamine maleate tablets	Taiyang Pharmaceutical Co., Ltd., Beijing, China	39
21		Shanxi Taiyuan Pharmaceutical Co., Ltd., Taiyuan, China	45
22		Shanxi Xinxing Pharmaceutical Co., Ltd. Linfen, China	36
23		Guangdong Nanguo Pharmaceutical Industry, Zhanjiang, China	39
24		Henan Jiushi Pharmaceutical Co., Ltd., Huixian, China	94
25	cefuroxime axetil tablets	Baiyunshan Tianxin Pharmaceutical Co., Ltd., Guangzhou, China	56
26		Beit Pharmaceutical Co., Ltd., Chengdu, China	29
27		Zhijun Pharmaceutical Co., Ltd. Shenzhen, China. (0.125 mg)	27
28		Zhijun Pharmaceutical Co., Ltd. Shenzhen, China. (0.25 mg)	89
29		United Laboratories International Ltd. (Zhongshang branch), Zhongshang, China	37
Total			1721

**Table 2 sensors-21-01088-t002:** Experimental results under different training and test set partitions.

Train:Test	Precision	Recall	F1	Accuracy
9:1	0.994	0.993	0.994	0.994
8:2	0.994	0.990	0.992	0.992
7:3	0.996	0.989	0.993	0.993
6:4	0.993	0.988	0.991	0.992
5:5	0.992	0.987	0.990	0.990
4:6	0.908	0.905	0.907	0.907
3:7	0.893	0.861	0.877	0.861
2:8	0.816	0.854	0.834	0.854

**Table 3 sensors-21-01088-t003:** Accuracy of various multi-class classification algorithms.

Train:Test	Bi-GAN	PLS-DA	Linear SVM	RBF_SVM	k-NN	BP-ANN	DBN	SAE	CNN
9:1	0.994	0.957	0.943	0.923	0.846	0.910	0.933	0.945	0.991
8:2	0.992	0.950	0.902	0.946	0.853	0.906	0.912	0.923	0.981
7:3	0.993	0.944	0.922	0.925	0.904	0.897	0.900	0.914	0.987
6:4	0.992	0.933	0.929	0.917	0.811	0.883	0.923	0.910	0.979
5:5	0.990	0.926	0.922	0.902	0.798	0.878	0.912	0.804	0.963
4:6	0.907	0.911	0.909	0.850	0.823	0.828	0.863	0.813	0.910
3:7	0.861	0.908	0.889	0.852	0.743	0.795	0.894	0.781	0.872
2:8	0.854	0.809	0.795	0.797	0.741	0.816	0.832	0.778	0.845

**Table 4 sensors-21-01088-t004:** Training and inferencing time of each algorithm (in second, training time/inferring time).

Train:Test	Bi-GAN	PLS-DA	Linear SVM	RBF-SVM	k-NN	BP-ANN	DBN ^1^	SAE ^2^	CNN ^3^
9:1	21.324/0.020	0.585/0.004	0.933/0.271	3.793/0.330	0.080/0.142	18.211/0.002	231.729/0.015	9.843/0.027	23.790/0.005
8:2	21.082/0.029	0.522/0.008	0.787/0.476	3.183/0.595	0.066/0.267	16.779/0.003	204.728/0.029	10.319/0.044	21.368/0.007
7:3	21.433/0.041	0.425/0.011	0.668/0.674	2.700/0.856	0.050/0.369	13.352/0.004	184.345/0.038	9.536/0.061	21.066/0.011
6:4	21.210/0.062	0.340/0.014	0.583/0.791	2.217/0.998	0.041/0.458	8.924/0.006	159.686/0.046	9.967/0.094	21.385/0.012
5:5	20.552/0.067	0.263/0.017	0.433/0.886	1.760/1.052	0.027/0.622	13.784/0.006	140.062/0.066	9.960/0.122	22.253/0.015
4:6	20.496/0.087	0.221/0.022	0.318/0.901	1.186/1.090	0.020/0.604	19.103/0.008	119.054/0.086	8.898/0.136	42.056/0.019
3:7	20.992/0.100	0.180/0.026	0.218/0.821	0.862/0.983	0.014/0.681	12.730/0.010	93.467/0.090	10.028/0.151	36.602/0.023
2:8	20.064/0.117	0.125/0.030	0.123/0.724	0.427/0.827	0.008/0.563	12.947/0.010	73.280/0.184	10.985/0.189	43.297/0.022

^1^ DBN’s RBM training is using the CPU algorithm, not the GPU algorithm, so its training time is very long. ^2^ SAE uses early stopping, so epoch training is uneven. ^3^ In CNN training, from 4:6, epochs are extended from 200 to 500, so the training time is longer.

## Data Availability

Not applicable.

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
