# Peer review of "Identification of Multi-Class Drugs Based on Near Infrared Spectroscopy and Bidirectional Generative Adversarial Networks"

_sensors, 2021, doi:10.3390/s21041088_

Round 1
Reviewer 1 Report
The manuscript provides a Bidirectional Generative Adversarial Networks for the analysis of NIR spectra collected on diverse drugs finalized to their classification.
The manuscript is interesting and it provides some novelties. I think it fits the scope of the journal and it could be suitable for publication after minor corrections.
I suggest to the author to clarify if the accuracy reported in the tables is calculated on the external set, or it is validated by internal validation.
Figure 2: There is a typo in the label on the y axes. Please replace "Sanples" by "Samples".
I am aware of one review focused on chemometrics applied to pharmaceutical data, I think it could be useful for readers to deepen some aspects, so the authors could add it in the references:
Biancolillo, A.; Marini, F. Chemometric Methods for Spectroscopy-Based Pharmaceutical Analysis. Front. Chem. 2018, 6 , 576
Reviewer 2 Report
Please identify the peaks in figure 4
Author Response
Thank you for review our papers and give us the high evaluation for us.
However, because our method uses deep learning to extract features automatically, it does not need to label the peaks. Comparison methods (PLS-DA, SAE, etc.) also do not need to use peak correlation features to identify.
At the same time, because each sub image represents 10-140 spectra, and there is baseline offset between the spectra, if the peaks is marked, each sub image will be clear only when the size of the whole image is expanded, and it will take at least three pages to describe clearly. This should be unnecessary.
We sincerely apologize for the inconvenience and hope to get your understanding and forgiveness.
Reviewer 3 Report
The aim of this paper was to develop a procedure for identification the true source of drugs based on NIR spectra and Bi-GAN chemometric classification. However, the paper does not show any example, how it to do. Valuable information from the practical viewpoint is that “the optimal accuracy rate is even more than 99%” – nothing else. Moreover, what does it mean “identification of multi varieties” of drugs?
In my opinion the paper should be published in a journal specializing in chemometry, especially in artificial neural network. The text is difficult to read due to specialized terminology. Only abbreviations are used in the case of multivariate statistical algorithms, such as PLS-DA, linear SVM, RBF SVM, k-NN, BP-ANN. For these reasons, this paper will be unreadable to the most of readers.
Other comments:
Table 1, line 28 – what does in mean “nzhen”?
The sentences should not be started with abbreviations or numbers, for example, see lines 109, 228, 275, 277,
More than ten references are proceedings of the conferences. I agree with the authors that proceedings of the conferences have sometimes important findings, however, they are inaccessible for those scientists who did not participate in these conferences. Thus, their usability as a reliable source of scientific information is generally very low.
Ref. 16 – journal name is missing, refs. 15, 20 – volume and pages numbers are missing.
Round 2
Reviewer 3 Report
All my comments have been adequately addressed in the revised manuscript. Moreover, the revised paper was improved a lot in relation to the previous version. I do not have any comments.